# Real space manifestations of coherent screening in atomic scale Kondo lattices

María Moro-Lagares[1,2,3], Richard Korytár[4], Marten Piantek[1,5], Roberto Robles [6], Nicolás Lorente[7,8], Jose I. Pascual[1,9,10], M. Ricardo Ibarra[1,5] & David Serrate [1,5,11]

The interaction among magnetic moments screened by conduction electrons drives quantum phase transitions between magnetically ordered and heavy-fermion ground states. Here, starting from isolated magnetic impurities in the Kondo regime, we investigate the formation of the finite size analogue of a heavy Fermi liquid. We build regularly-spaced chains of Co adatoms on a metallic surface by atomic manipulation. Scanning tunneling spectroscopy is used to obtain maps of the Kondo resonance intensity with sub-atomic resolution. For sufficiently small interatomic separation, the spatial distribution of Kondo screening does not coincide with the position of the adatoms. It also develops enhancements at both edges of the chains. Since we can rule out any other interaction between Kondo impurities, this is explained in terms of the indirect hybridization of the Kondo orbitals mediated by a coherent electron gas, the mechanism that causes the emergence of heavy quasiparticles in the thermodynamic limit.

[1] Laboratorio de Microscopias Avanzadas, Instituto de Nanociencia de Aragón, University of Zaragoza, E-50018 Zaragoza, Spain. [2] Institute of Physics, Academy of Sciences, Prague 16200, Czech Republic. [3] Regional Centre of Advanced Technologies and Materials, Faculty of Science, Department of Physical Chemistry, Palacky University, Olomouc 78371, Czech Republic. [4] Department of Condensed Matter Physics, Faculty of Mathematics and Physics, Charles University, 121 16 Prague 2, Czech Republic. [5] Dpto.Física Materia Condensada, University of Zaragoza, E-50009 Zaragoza, Spain. [6] Catalan Institute of Nanoscience and Nanotechnology (ICN2), CSIC and BIST, Campus UAB, Bellaterra, 08193 Barcelona, Spain. [7] Centro de Física de Materiales CFM/MPC (CSIC-UPV/EHU), 20018 Donostia-San Sebastián, Spain. [8] Donostia International Physics Center (DIPC), 20018 Donostia-San Sebastian, Spain. [9] CIC NanoGUNE, E-20018 Donostia-San Sebastián, Spain. [10] IKERBASQUE, Basque Foundation for Science, E-48011 Bilbao, Spain. [11] Instituto de Ciencia de Materiales de Aragón, CSIC - Universidad de Zaragoza, 50009 Zaragoza, Spain. Correspondence and requests for materials should be addressed to D.S. (email: serrate@unizar.es)

The interaction of a localized magnetic moment with a metallic host gives rise to the Kondo many-body state (KS) [1,2]. The Kondo temperature defines the characteristic energy scale of the single-ion KS, $k_B T_K$. For $T \lesssim T_K$, the conduction electrons align antiferromagnetically to the localized moment, resulting in a complete magnetic screening of a spin-1/2 impurity, and forming a singlet in the KS (Fig. 1b). In this way, the KS can be understood as the coupling of the quantum numbers of the conduction electrons and the localized moment to form a hybrid quasiparticle.

A Kondo lattice is a set of localized magnetic impurities arranged in a regular pattern, which interacts with a bath of delocalized conduction electrons. When the two subsystems are weakly coupled, the Kondo lattice falls in the paramagnetic or antiferromagnetic (AFM) regimes (Fig. 1a, c), both characterized by a finite impurity spin. The AFM coupling is driven by the Ruderman-Kittle-Kasuya-Yosida (RKKY) indirect exchange interaction. In the limit of a large coupling strength, $J\rho$, between localized moments and extended states of the conduction electrons, the two subsystems cannot be treated separately. The Kondo lattice introduces a new energy scale, $k_B T^*$, which plays the role of $T_K$ in the sense that below $T^*$ the magnetic susceptibility starts being anomalously reduced due to partial screening[3]. But $T^*$ and $T_K$ are not trivially related[4]. In an extended Kondo lattice, for $T \ll T^*$, the system falls in the heavy Fermi liquid (HFL) phase (Fig. 1d). The HFL state is non-magnetic, because the conduction electrons fully screen the lattice spins. As in the case of the single-ion KS, the localized magnetic states contribute to the quasiparticle band developed by the coupled Kondo states,

becoming in this way a part of the Fermi volume or, equivalently, delocalized in space[5]. For this reason, this process is often called spin de-confinement.

The Kondo screened state and the AFM inter-impurity order are separated by a quantum critical point (QCP, Fig. 1c, d)[4,6–8], which makes the ground state very sensitive to external parameters such as temperature, pressure or magnetic field. Using scanning tunnelling microscopy (STM) and spectroscopy (STS), the atomistic perspective of this transition has been studied in dimers[9–16]. It was found that the KS develops magnetic correlations for sufficiently small inter-impurity distances ($d$), indicating that the dimers are in the range of small $J\rho$, where the HFL phase cannot set in (Fig. 1). Since $J\rho$ cannot be tuned in situ, the HFL state could not be addressed in this kind of atomic dimers. Finite size lattices composed by spin moments of molecular self-assemblies were also reported, but here again the AFM coupling dominates over Kondo screening due to either RKKY[17] or superexchange[18] coupling.

Here, we fabricate chains of coupled single-ion Kondo resonances by atomic manipulation in order to characterize in real space the crossover between the KS and the finite size analogue to a HFL (Fig. 1b, d). To this end, we choose a sample with a strong $J\rho$ coupling, so that Kondo screening overcomes any intersite interaction. The 3d orbitals of individual Co atoms adsorbed on a clean Ag(111) surface play the role of localized spins and metal host, respectively. Although Co on Ag(111) displays a similar Kondo strength as on Cu(111) and Au(111)[19], the near Fermi level electronic structures of these surfaces are markedly different. In particular, the onset of the surface conduction band of Ag(111)

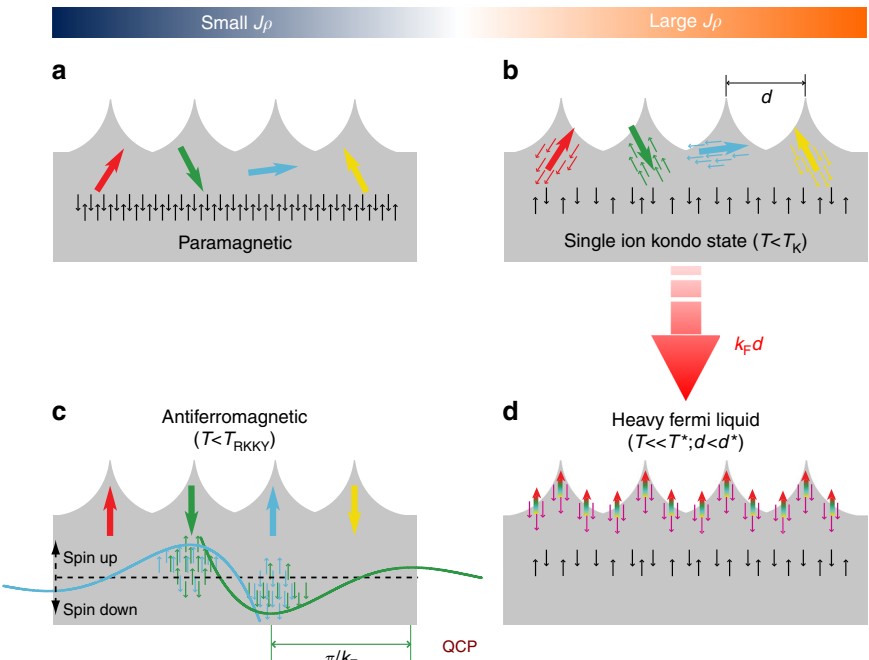

**Fig. 1** Schematic phase diagram of a Kondo lattice. The four possible phases of a Kondo lattice: **a** Paramagnetic (PM) regime. **b** Single-ion Kondo state (KS) with non-interacting impurities. **c** Magnetically ordered state via RKKY interaction. **d** Heavy Fermi liquid (HFL) phase. $J$ is the exchange coupling between the local magnetic states (big arrows) and the Fermi gas (occupying the grey regions), and $\rho$ the density of states of the conduction electron bath around the Fermi level. The small black arrows represent the spins of a regular Fermi gas decoupled from the impurity spins, whereas the small coloured arrows represent the spin of conduction electrons coupled to the magnetic impurities. The role of the inter-impurity distance ($d$) is described by the red wide arrow representing the dimensionless parameter $k_F d$, where $k_F$ is the Fermi wave vector of the conduction electrons. A quantum critical point (QPC) appears at the boundary between the antiferromagnetic (AFM) and the HFL states as a function of $J\rho$. For small coupling (small $J\rho$), the KS or the PM state can be driven to the AFM state by decreasing $d$ or the temperature. For large coupling ($J\rho$ to the right of the QCP), the KS can be driven to the HFL phase by decreasing $d$, as we do in this work. Note that in the HFL state, all impurity spins are coherently screened by all conduction electrons participating in the heavy fermion band, in contrast to the single-ion KS where each impurity spin is individually screened. In the AFM state, the spin polarization of the the electron bath induced by the green and blue impurity spins is represented by a decaying oscillatory function (responsible for the RKKY exchange coupling)

is right below the Fermi level, which gives rise to strong fluctuations of the density of screening electrons and an anomalously large Fermi wavelength[20]. By means of an analysis of the multi-impurity Anderson model (MIAM), we link the Ag(111) surface electronic structure to the experimental fingerprints of coherent and collective screening of the artificial Kondo lattice. We probe experimentally the de-confinement of Co magnetic moments upon formation of a collective KS by imaging the amplitude of the Kondo resonance in tunnelling conductance spectra, which delocalizes away from the position of the impurity centres. The delocalization occurs in a gradual manner as the inter-impurity distance in dimers and chains is reduced, which is a way of tuning the ratio $T^*/T_K$. Note that we use the lattice geometry as the control parameter to induce collective screening right from the single-ion KS. Such transitionis not accessible in temperature dependent studies of spectral features in extended Kondo lattices[21–24], because the large impurity density impedes the formation of a pure single-ion KS.

## Results

**Single-ion Kondo state**. The KS produces a resonant peak in the density of states (DoS) around the Fermi level, which manifests in STM as a zero-bias feature (ZBF) in differential conductance d$I$/d$V$ spectra[25,26]. On metals, the ZBF can be generically described by a Fano lineshape[25,27–29]. Figure 2a shows the fit of the ZBF measured over a single Co atom to a Fano function. We describe the resonance as dI/dV($\epsilon$) = $\rho_0(\epsilon)\rho_K(\epsilon)$, where $\rho_0$ is a

polynomial background and the Fano function enters $\rho_K$ linearly (see Supplementary Note 1):

$$\rho_K(\epsilon) = A_0 + A_K \frac{(q + \xi)^2}{1 + \xi^2} \text{ with } \xi = \frac{\epsilon - \epsilon_0}{\Gamma_0/2} \quad (1)$$

Here $\epsilon = eV$, $V$ the sample bias voltage, $A_K$ the Kondo resonance amplitude, and $\Gamma_0$ the resonance linewidth. Care must be taken that there are not tip resonances contributing to the ZBF, which could affect the intrinsic values of the fitting parameters (see Supplementary Fig. 2). $\Gamma_0$ is related with the Kondo temperature as $2k_B\sqrt{(\pi T)^2 + 2T_K^2} = \Gamma_0$ if $k_BT/\Gamma_0 \to 0$[30] (Supplementary Note 1). We obtain $k_BT_K = 3.2 \pm 0.6$ meV for isolated single atoms far from defects and impurities[20]. This, together with the negligible unixial aniostropy energy of 0.14 meV compared with $k_BT_K$ that we obtained by density functional theory (DFT) calculations (see Methods), is consistent with a fully screened KS at the working temperature ($k_BT = 0.09$ meV).

**Interaction of two Kondo states**. We are interested in a dense system, with interatomic distances below the length scale in which the single-ion KS picture does not suffice to describe the Physics. To determine such characteristic length scale, we probe variations in the Kondo ZBF when a second Co atom is approached (Supplementary Fig. 3). The interatomic distance, $d$, is expressed as a multiple of the Ag(111) lattice parameter along

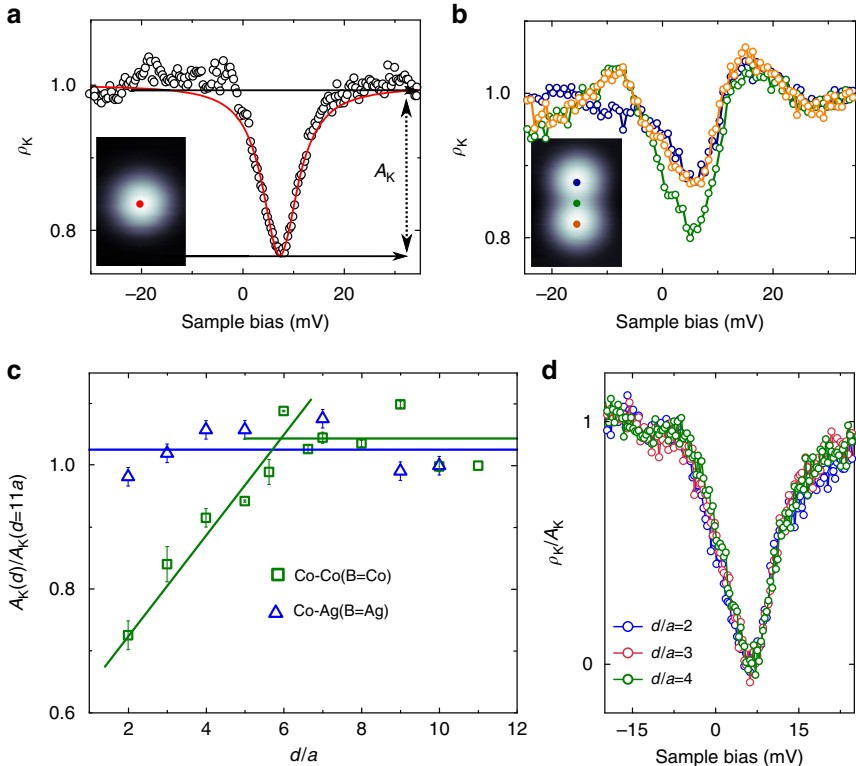

**Fig. 2** Kondo resonance of individual Co atoms and dimers on Ag(111). **a** d$I$/d$V$ at the atom centre (circles), and fit to Eq. (1) yielding $\Gamma_0 = 8.6 \pm 0.6$ meV, $q = 0.02 \pm 0.02$ and $\epsilon_0 = 7.1 \pm 0.2$ meV (red line). **b** d$I$/d$V$ over Co atoms (blue and orange open circles) in a dimer at $d = 2a$ ($a = 2.89$ Å is the Ag(111) surface lattice parameter) and over the midpoint between them (green circles). Insets display the corresponding constant current topographic images (width 1.3 nm) and the dots mark the position where spectroscopy was taken. **c** $A_K$ (defined in panel **a**) variation of a Co atom as another Co (squares) or Ag (triangles) atom is approached. Data are extracted from fits to individual spectra acquired over the atoms, and the amplitude is normalized by its corresponding value with the second atom at a distance of $d = 11a$ from the fixed Co (See Supplementary Note 2). Error bars are standard deviations obtained during the fitting procedure. **d** d$I$/d$V$ normalized by $A_K$ showing that the shape of the ZBF does not change as Co atoms are approached (i.e., $\Gamma_0$, $q$ and $\epsilon_0$ remain constant). In panels **a**, **c**, and **d**, the set point is the general value given in Methods, in panel **b** the set point is $-20$ mV and 15 pA, with lock-in modulation of 1 mV rms

the [1$\bar{1}$0] close packed directions, $a = 2.89$ Å. As shown in Fig. 2c, the Kondo amplitude measured on top of the Co atom drops gradually as the separation decreases from $d = 6a$ to $d = 2a$, whereas the lineshape (given by $q$ and $\Gamma_0$) of the ZBF remains constant (see Fig. 2d and raw data in Supplementary Fig. 4) within our experimental uncertainty ($q = 0.16 \pm 0.1$ in this dataset).

For the compact dimer ($d = a$), the ZBF is totally suppressed in the d$I$/d$V$ spectrum (Supplementary Fig. 5). This distance is within the regime where orbital overlap can occur, triggering direct magnetic exchange that leads to a new entangled spin spin state[8–10,31], which presumably is not compatible with the Kondo effect. On the other hand, indirect RKKY coupling of the order of $k_B T_K$ does not suppress the ZBF, but it is known to have a stark effect on it, producing either a splitting[13,17,32–34] or a broadening[11,12,14,15] of the resonance. In contrast, the lineshape ($q$, $\Gamma_0$) of the ZBF does not change for $d \geq 2a$ in our Co dimers. Consequently, we conclude that there are not direct or indirect magnetic interactions, in line with the negligible exchange coupling of Co atoms on the surface at $d = 2a$ calculated by DFT ($|\Delta E_{ex}| = 0.64$ meV $\ll k_B T_K$). In the context of Kondo lattices, this implies that our system lies in the region of large $J\rho$ of the phase diagram shown in Fig. 1, where Kondo screening overcomes indirect RKKY magnetic coupling. Note that there are other interactions with the immediate environment that would also produce noticeable changes of the lineshape, which can thereby be discarded also for our Co chains. Examples are magnetocrystalline anisotropy[35], shift of atomic states[36,37], adsorption geometry[38,39], chemical bonding[16,18], or charge doping[40].

Still, $A_K$ decreases smoothly in a distance range ($d \leq 5a$) well above the spatial extent of Co atomic orbitals, pointing out that some type of indirect interaction (other than RKKY) mediated by the conduction electrons comes into play. More interestingly, in the dimer at $d = 2a$, the amplitude of the ZBF is larger at the midpoint between the Co atoms than over their centres (Fig. 2b). This suggests that the observed $A_K$ decrease is not simply reflecting an overall weakening of the KS, but rather a spatial redistribution of its associated many-body DoS. In order to study the spatially resolved intensity of the KS wave function, we constructed $A_K(x, y)$ maps from constant height d$I$/d$V$ images at the two applied biases marked in Fig. 2a. These measurements are performed at constant height, and thereby our data are free of feedback artefacts, in such a way that the $A_K$ image is totally independent from the DOS in the absence of Kondo fluctuations, and portrays exclusively the Kondo-related intensity (for further details on the method see Supplementary Note 1 and Supplementary Fig. 1). The map of the single atom is shown in Fig. 3a, b, yielding a circular region with non-vanishing $A_K$ of diameter $\simeq 5a$ around the centre.

Figure 3c compares constant height tunnelling current images and $A_K(x, y)$ maps of Co dimers for several $d$ values. In the most diluted case ($d = 5a$), each atom possesses an individual Kondo resonance located right at the atom centre. As $d$ decreases, the $A_K$ intensity deviates from the atomic cusps and shifts towards the centre of the dimer. At $d = 2a$ (dense regime), the two KS collapse in a single one with maximum $A_K$ in between the atoms and approximately halved intensity at the atoms position, as shown by the horizontal profiles in Fig. 3d. This redistribution of the Kondo amplitude explains the decrease of $A_K$ for atom pairs closer than $d = 5a$. Furthermore, it suggests that the screening of the magnetic moments is a collective effect involving the two atoms and the metallic host. Finally, the amplitude redistribution observed for Co dimers is genuinely due to an interaction between two single-ion KS, because approaching a non-magnetic Ag atom to a KS does not give rise to any systematic variation of $A_K$ (Fig. 2c). In addition, the $A_K$ map of the Ag-Co dimer at $d = 2a$ (Supplementary Fig. 6) is the same as for the isolated Co.

We exclude interference between tunnelling channels through both atoms as responsible for the enhanced $A_K$ signal when the tip is at the middle point. First, the experimental measure of interference is the $q$-factor of the Kondo dip, which together with $\Gamma_0$ remains constant across the dimer in our fits to Eq. (1). Second, we have confirmed that the intensity redistribution towards the dimer centre is more intense the sharper the tip is (sharpness is quantified by the corrugation of the tunnelling current profiles at $V_{ref}$ in dimers at $d = 2a$).

**Kondo redistribution in finite Kondo lattices**. In order to find out how these results translate into a periodic Kondo lattice, we have built atomic chains of $N$ atoms length ($3 \leq N \leq 14$) with spacing of either $d = 2a$ or $d = 3a$. Selected images of $A_K$ and tunnelling conductance are shown in Fig. 4. The resonance width ($\propto T_K$) of an atom remains constant as the chain is constructed sideways, as well as in all atoms within a chain of arbitrary length (see Supplementary Fig. 7). This indicates again that there is no magnetic exchange among impurities, in contrast to the case of a molecular the Kondo chain[18]. For all values of $N$ at $d = 3a$, Fig. 4a exhibits a set of peaks in $A_K(x, y)$ coinciding with the atoms positions. However, when the interatomic distance is reduced to $d = 2a$ in Fig. 4b, the Kondo amplitude spreads over the chains and displays only a weak modulation (below the experimental uncertainty) along all inner atoms. The difference between $d = 2a$ and $d = 3a$ is consistent with our observation on the dimers. In addition, the end atoms of the chains show a striking feature. For the chains with $N > 4$ and $d = 2a$, the two outermost end atoms display a prominent edge lobe of $A_K$ with maximum signal in between them. Enhanced edge lobes in the $A_K$ intensity are also found in chains with $d = 3a$, although in this case they coincide with the end atoms, and disappear for $N = 10$ (Fig. 4a).

**The MIAM**. To rationalize the Kondo redistribution pattern, we employ a MIAM where each Co atom is represented by a single Anderson impurity (d-state), which hybridizes with a single band of conduction electrons (k-states), see Methods. We include hopping terms between each k-state and any of the $i = 1, ..., N$ impurities, which give rise to a hybridization function $\Gamma_{ij}(\omega)$, a matrix in the impurities sub-space. The quantity $\Gamma_{ij}(\omega)$ accounts entirely for the effect of the conduction band on the impurities. In the single impurity limit, $\Gamma_{11}(\omega = 0)$ determines the value of $T_K$ along with $U$, $\epsilon_d$ and the bandwidth (see Methods). In the multi-impurity case, the off-diagonal terms of $\Gamma_{ij}(\omega)$ represent hybridization between $i$ and $j$ impurities mediated by the conduction band. We disregard in the MIAM a Heisenberg term between impurity spins because our low bias d$I$/d$V$ spectra are not compatible with any kind of magnetic exchange interactions (see Fig. 2d, and Supplementary Figs. 4 and 7). We calculate the impurity Green's function $G_{ij}(\omega)$ by a self-consistent many-body perturbation theory (see Supplementary Note 3) and obtain the local spectral function $A_{imp/subs}(\mathbf{r}, \omega)$ of the impurities/substrate (representing the local DoS) as explained in Methods.

We first confirm that the delocalization of the Kondo amplitude in two-impurity systems is supported by our MIAM calculations. To this end, we set the impurity parameters ($\epsilon_d$, $U$) and $\Gamma_{ii}(0)$ into the Kondo limit and plot $A_{imp}(\mathbf{r}, \omega_0)$ maps of dimers for varying $\Gamma_{12}(0)$, with $\omega_0$ the energy at which the Kondo resonance is centred. For independent impurities, i.e., when $\Gamma_{12} = 0$, the map displays an incoherent sum of the two Co wave functions ($\varphi_i$ in Eq. (6)) used to reproduce the experimental $A_K$ for the single atom case (cf. Figs. 3a and 5a). In the opposite limit of strong cross-talk between impurities, we set $\Gamma_{12} = \Gamma_{11} = \Gamma_{22} = \Gamma_{21}$. In line with the experimental results of Fig. 3c, the spectral function map is now a single cloud with maximum amplitude at

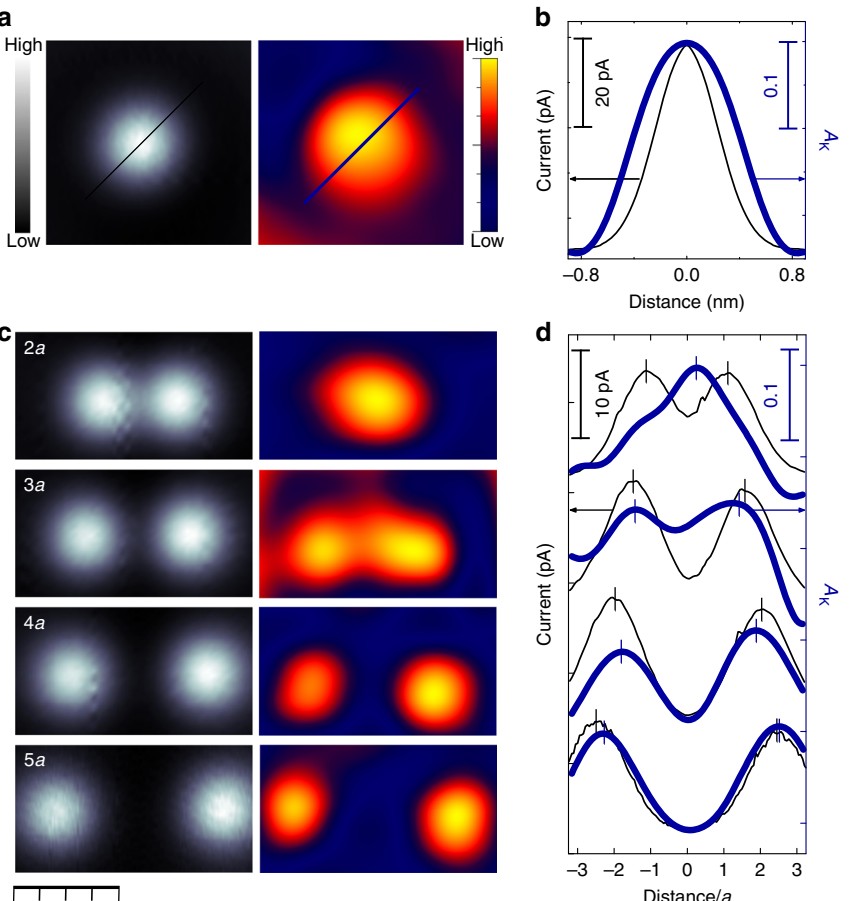

**Fig. 3** Mapping of the experimental Kondo resonance amplitude in Co dimers. **a** Simultaneous constant height images of the tunnelling current (left) and Kondo amplitude, $A_K$ (right) of an individual Co atom. $A_K$ is the depth of the Kondo resonance, see Eq. (1) and Fig. 2a. **b** Line profile across the atom (see lines in (**a**)) of the current (thin black line) and $A_K$ (thick blue line). **c** The same as (**a**) for a Co dimer at $d = 2a$, $3a$, $4a$, and $5a$ modified by atomic manipulation ($a = 2.89$ Å). **d** Horizontal profiles of the tunnelling current (black thin lines), and $A_K$ (blue thick line) from (**c**). Data were vertically offset for clarity. All tunnelling current images (greyscale) were acquired at $V_{ref} = 17$ mV sample bias. Bottom scale bar corresponds to 1 nm

the dimer centre (Fig. 5b), reflecting a coherent superposition of the original Kondo clouds.

For longer chains, we express explicitly the dependence of $\Gamma_{ij}(\omega)$ on the distance between impurities $i$ and $j$, which for our evenly spaced chain amounts to $\delta r = d|i - j|$. We assume that the inter-impurity hybridization is mediated by a translationally invariant substrate. Then, the asymptotic distance dependence of $\Gamma_{ij}(\omega)$ must be that of a spherical wave-function emanating from one impurity and reaching the other. Its envelope, for a coherent $D$ dimensional Fermi gas, should get reduced by the factor $\delta r^{-(D-1)/2}$ and, owing to the Bloch phase picked at the Fermi surface, it should oscillate periodically with the Fermi wavelength ($\lambda_F$). The impurities are now considered point-like, and the analytic expressions of $\Gamma_{ij}(\omega)$ in $D = 2$ and $D = 3$ for a homogeneous electron gas with bandwidth $2E_0$ are, as derived in the Supplementary Note 3:

$$\Gamma_{ij}(\omega) = \Gamma_0 \, J_0(k_F d|i - j|) \cdot \Theta(E_0^2 - \omega^2) \quad \text{in } D = 2 \quad (2)$$

$$\Gamma_{ij}(\omega) = \Gamma_0 \, \frac{\sin(k_F d|i - j|)}{k_F d|i - j|} \cdot \Theta(E_0^2 - \omega^2) \quad \text{in } D = 3 \quad (3)$$

Apart from the dimensionality, the only remaining parameter is the dimensionless product $k_F d$, where $k_F = 2\pi/\lambda_F$. As in the two-impurity case, we tune the system parameters into the Kondo limit. We analyze the change of the local spectral function of the

conduction electrons due to the impurities, $\delta A_{subs}(\mathbf{r}, \omega_0)$. We begin with $D = 2$ and $k_F d = 0.18\pi$, very close to the experimental scenario with $d = 2a$ and $k_F = 0.085$ Å$^{-1}$ [41]. Selected maps of $\delta A_{subs}(\mathbf{r}, \omega_0)$ are shown in the insets of Fig. 5c for $N = 9$, 12, and 15. For the shortest chains up to $N \leq 9$, the map consists of a single cloud. In the range $10 \leq N \leq 14$, two distinct maxima appear at the edges of the chain. For even longer chains, a new peak emerges in the chain centre merging with the edge features. Remarkably, the source of the edges and other patterns can be traced to the off-diagonal elements of the impurity Green's function $G_{ij}(\omega)$ (see Supplementary Fig. 9). In contrast to $D = 2$, we do not observe the edge enhancement for $D = 3$. This is due to a faster decay of the substrate-mediated hybridization across the chain. Hence, the presence of edge lobes is a fingerprint of the $D = 2$ character of the conduction band, which suggests that the 2D Shockley surface state governs the substrate-mediated impurity hybridization. This is in agreement with recent experimental and theoretical studies demonstrating that the surface state plays a primary role in the Kondo effect of Co/Ag(111) [20,42].

The role of $\lambda_F$ is neatly demonstrated by the general behaviour observed for various impurity spacings $d$. Figure 5c shows the chain lengths for which the two edge lobes exist as a function of $d$, both theoretically and experimentally. Upon increasing $N$, the edge lobes appear, then remain up to a critical length, and finally are replaced by more structured patterns in very long chains. Remarkably, we observe theoretically a robust scenario in which

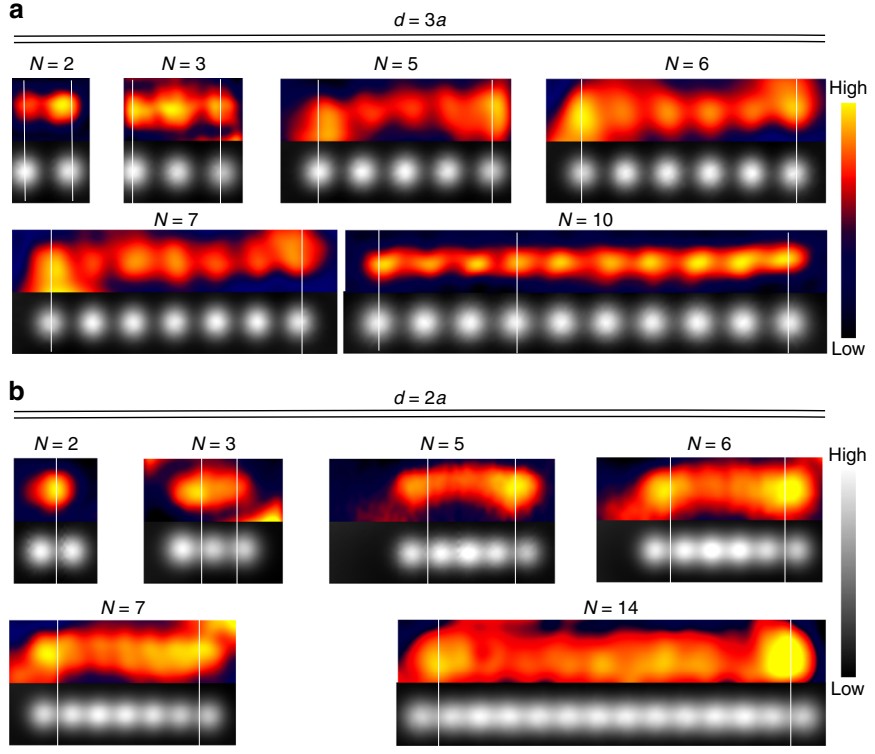

**Fig. 4** Spin de-confinement in 1-D artificial Kondo lattices. Simultaneous constant height tunnelling current (greyscale) and Kondo amplitude $A_K$ (red-yellow scale) maps of $N$ Co atoms chains with spacing $d = 3a$ (**a**) and $d = 2a$ (**b**). White lines are guides to highlight the position of the $A_K$ maxima with respect to the atoms' position. **a** At $d = 3a$, all atoms retain an associated Kondo resonance, with maximum $A_K$ at end atoms for $N < 10$. **b** At $d = 2a$, the $A_K$ edge maxima are found in between the two end Co atoms of each side. The bottom scale bar represents 2 nm and applies to the entire figure. All tunnelling current images (greyscale) were acquired at $V_{ref} = 17$–$20$ mV sample bias

the edge lobes show up in chains with lengths of at least $L \approx 0.75\lambda_F$ and merge when $L \gtrsim 1.1\lambda_F$. Thus, the control parameter is not the spacing, but the total length $L = d(N - 1)$. The scattering of the boundary region is just $d$, i.e., one impurity of the chain. We note that the Kondo resonance needs to be fully developed ($T < T_K$) since, by increasing the temperature, the patterns get weaker and disappear.

## Discussion

For non-interacting impurities, the spectral function $A_{imp}$ (see Fig. 5a) reproduces quite accurately the circular objects described by the Kondo amplitude that are observed experimentally (Fig. 3a, b). When the substrate-mediated inter-impurity hybridization $\Gamma_{12}(\omega)$ is turned on, the spectral function displays the same delocalized pattern as in the experimental Kondo amplitude of the dimer at $d = 2a$ (cf. Figs. 3c and 5b). Note that $\Gamma_{12}(\omega)$ originates from coherent hoppings of electrons from a single conduction band to either Anderson impurity. The fact that the impurity spins are more coupled to conduction electrons away from the magnetic orbitals is the signature of spin de-confinement[43]. The latter has been attributed to the intersite RKKY interaction in ref. [17]. As we show, the spin de-confinement in Co dimers is linked to the effect of the overlap of Kondo clouds. Formally, this is expressed by $\Gamma_{12}(\omega)$, which is second order in $V_{ki}$, whereas the RKKY appear only in the fourth order in $V_{ki}$ (see Supplementary Note 4).

Further evidence of collective screening is provided by the comparison of $A_K$ maps of the long chains (Fig. 4) and its closest theoretical analogue, $\delta A_{subs}$. As a result of the coherent interference of the electron gas coupled to the impurity lattice, the

theory predicts a critical chain length, $L_c$, for the appearance of Kondo amplitude maxima near the edge atoms, which scales with the Fermi wavelength of the conduction band. Moreover, it also predicts the suppression of the edge lobes above a maximum length, $L_M$. As illustrated in Fig. 5c, both predictions are experimentally observed in chains with two different spacings $d = 2a$ and $3a$. The model approximations lead to quantitative differences in the threshold lengths: $L_{c-M}/\lambda_F = 0.75$–$1.15$ and $0.33$–$1.06$ for theory an experiment, respectively. Despite of this, the excellent qualitative agreement between the experimental $A_K$ maps and the MIAM calculation implies that our Co atoms cannot be viewed as a chain of local scatterers, because they give rise to a non-local Kondo resonance. The non-local origin of the conductance patterns in the Kondo lattices constructed in this work contrasts with the patterns found in quantum mirages, which were explained by purely local scattering[44]. Non-local Kondo screening has also been observed in $Mn_xFe$ chains[31], where the spin of the constituent atoms is strongly entangled by exchange coupling. Here we explore the opposite limit with negligible coupling between the atomic spins of the chain. The $Mn_xFe$ chains act as a single Kondo impurity with a complex pattern of Kondo fluctuations spreading over several atoms in the chain, whereas for Co/Ag(111) chains we observe manifestations of collective screening of a set of individual impurities.

The single-ion picture clearly fails for our chains and dimers with the shortest impurity spacings, which suggests a substantial increase in $T^*$[3,4] above the working temperature of 1.1 K when $d$ decreases below $3a$. In this regime, we observe the defining characteristics of the HFL phase in a Kondo lattice[2,5]: spin de-confinement and collective screening. The HFL state has been investigated by STM in 2D[43] or 3D[22–24] dense self-assembled

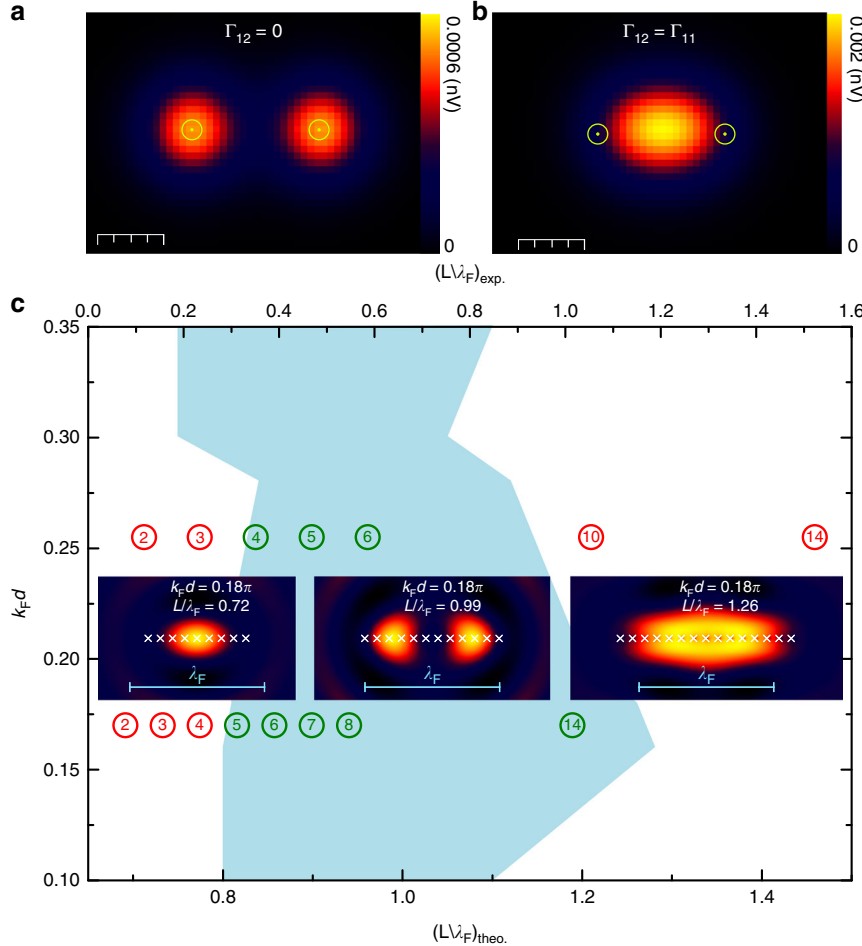

**Fig. 5** Spectral function maps of the MIAM in the Kondo regime. **a**, **b** Theoretical map of the impurities' spectral function evaluated at the energy of the Kondo peak of a dimer with spacing $d = 2.6a = 7.5$ Å. Scale bars represent 4 Å. The two figures (**a**, **b**) differ only in the off-diagonal element of the hybridization function ($\Gamma_{11} = 10$ meV, $\Gamma_{12} = 0$ and $\Gamma_{11} = \Gamma_{12} = 10$ meV, respectively). The other model parameters are given in Methods. The impurities' positions are marked by yellow circles. **c** Diagram for the appearance of edge lobes in spectral function map of conduction electrons (see Methods, $\Gamma_0 = 60$ meV), constructed by analyzing chains of different spacing $d$ and length $L = (N-1)d$. The pale blue area corresponds to the $d - L$ region where edge lobes appear. Insets represent selected spectral function maps of chains with $N = 9$, 12, 15 atoms and spacing $d = 2.2a$, exemplifying the presence or absence of edge lobes. The spectral function is again evaluated at the energy of the Kondo dip. The impurities' positions are marked by white crosses. The chain length for theoretical results is given in the bottom abscissa in units of $\lambda_F$. Green(red) empty circles are experimental data points from $A_K$ maps with (without) edge lobes in chains as deduced from $A_K$ maps in Fig. 4 for $d = 2a$ and $d = 3a$ (the images of the chains not shown there are available in Supplementary Fig. 8). The number enclosed by the circles is the number of atoms, $N$. The experimental chain length is given in the upper abscissa

Kondo lattices. In these examples, the transition between KS (independent impurity behaviour with energy scale $\sim T_K$) and the HFL state (energy scale $\sim T^*$) could not been reached, unlike the case of our Co chains built atom-by-atom where the origin of the Kondo features can be traced down to the single-ion state. The conditions to obtain both states in the same sample cannot be met either in the case of bulk crystalline heavy fermion compounds[3,4,45–47].

The possibility of observing the real space structures in the adatom Kondo lattice is intimately linked to the fact that the chains are finite. This grants access to quantum size effects of the Kondo lattice Physics. In exchange, some more common features of the Kondo lattice problem in bulk heavy fermion compounds are missing. For instance, the opening of a hybridization gap upon formation of the HFL phase[23,48,49], which should manifest as a new ZBF in the dI/dV spectra[21,22,50]. We do not observe any additional feature on top of the Kondo resonance upon the formation of the Kondo lattice. The reason for this absence remains an open question for further theoretical investigations: the finite size of the Co lattice, its coupling to a metallic host with lower

dimensionality, or even an eventual opening of the gap at temperatures lower than experimentally available.

To sum up, we have visualized in real space the onset of HFL behaviour as the redistribution of the Kondo amplitude (and consequently spin de-confinement). In dimers, it appears as a delocalization of the Kondo amplitude towards its centre. In chains, it manifests as patterns at length scales of the order of $\lambda_F$. Apart from the Kondo temperature, the theoretical model reproduces the essential experimental observations without adjustable parameters: the dimensionless product $k_F d$ is fixed by the experimental conditions, and the leading dimension of the electron gas is two. Therefore, our microscopic description should apply for different material parameters, and for arbitrary lattice site and dimensions, laying foundations to further investigations of correlated nanostructures engineered with atomic precision.

## Methods

**Experimental**. The experiments have been performed in a SPECS Joule-Thompson STM[51] in ultra-high vacuum (chamber base pressure $<1 \times 10^{-10}$ mbar) at 1.14 K. Some control experiments were repeated at 4.7 K. All images and spectroscopy data

are acquired at constant tip sample distance unless stated otherwise (feedback opened over the Ag(111) surface at regulation set point of $-100\,\text{mV}$ and 40 pA). In our set up, the tip is grounded and bias voltage is applied to the sample. The Ag (111) surface was cleaned by repeated cycles of $\text{Ar}^+$ sputtering at 1 kV, $1 \times 10^{-6}$ mbar and annealing at 450 °C. Co atoms were deposited in situ onto the Ag(111) substrate from an e-beam evaporator loaded with a high purity Co rod focused at the STM sample stage held at $T < 5\,\text{K}$ to prevent atoms diffusion. Ag atoms were extracted from the substrate by means of controlled tip indentation. We used lateral atomic manipulation to move individual atoms to designated positions and construct artificial atomic ensembles. Atom manipulation was performed exerting a lateral force on the atom with the STM tip in the attracting regime (pulling mode, manipulation parameters $I_0 \geq 40$ nA and $V_0 = 3$ mV). The tip is an electro-chemically etched W wire, which is in situ ion milled by means of $\text{Ar}^+$ sputtering at 5 kV. Afterwards, the tip apex is gently dipped in the Ag(111) surface until it becomes structurally stable against atomic manipulation. Co atoms can be adsorbed on the two types of hollow sites of the Ag(111) surface: fcc or hcp. Our data are fairly independent on the site type, with the exception that in one of them the resonance amplitude is systematically lower than in the other. In order to overcome this duality, we ensure that all atoms are in equivalent sites by carefully assembling all atomic structures along [1$\bar{1}$0] atomic closed packed directions of the Ag(111) substrate. All differential conductance d$I$/d$V$ data are obtained using a lock-in amplifier at 911 Hz and modulation voltage of 0.5 mV rms for spectra and 2 mV rms for imaging. The whole image processing has been carried out with WSxM[52].

**Ab initio calculations**. The electronic structure of Co/Ag(111) has been evaluated using the VASP-DFT code, including spin-orbit coupling. In the two types of hollow adsorption sites, we obtain a spin magnetic moment of 2.04 $\mu_B$ with negligible orbital moment and very small uniaxial anisotropy of 0.14 meV. Therefore, the leading interaction is Kondo screening. In absence of electronic correlations ($U = 0$), Co atoms exhibit two unoccupied antibonding d-orbitals and thereby they can be described by an isotropic $S = 1$ spin system. The inter-impurity exchange energy in the dimer was calculated as the energy difference between the parallel and anti-parallel spin alignment.

**Theoretical DoS in the MIAM**. The Hamiltonian of the MIAM reads

$$\hat{H} = \sum_{k\sigma} \epsilon_k \hat{c}^\dagger_{k\sigma} \hat{c}_{k\sigma} + \epsilon_d \sum_{m\sigma} \hat{d}^\dagger_{i\sigma} \hat{d}_{i\sigma} + U \sum_m \hat{d}^\dagger_{m\uparrow} \hat{d}_{m\uparrow} \hat{d}^\dagger_{m\downarrow} \hat{d}_{m\downarrow} + \\ + \sum_{k\sigma} \sum_m \left( V_{km} \hat{c}^\dagger_{k\sigma} \hat{d}_{i\sigma} + \text{h.c.} \right) \quad (4)$$

where the first term describes a single conduction electron band (substrate), the second term stands for impurities (adatoms) $m = 1, \ldots, N$ with equal on-site energy $\epsilon_d$, the third term is a Hubbard-type repulsion on each impurity orbital and the last term is impurity-substrate hybridization. We do not assume any direct inter-impurity hoppings. We take the approximation $U = \infty$ and calculate the multi-impurity Green's function $G_{ij}(\omega)$ by self-consistent many-body perturbation theory, see Supplementary Note 3 for details.

Upon integrating out the conduction electrons, the impurities acquire an embedding self-energy $\Sigma_{ij}(\omega)$. For non-interacting electrons ($U = 0$), the diagonal terms $\Sigma_{ii}$ shift and broaden the impurity levels and the off-diagonal terms provide substrate-mediated hybridization. The central role is played by the hybridization function $\Gamma_{ij}(\omega) = \frac{i}{2\pi} \left[ \Sigma_{ij}(\omega) - \Sigma^*_{ji}(\omega) \right]$, given by the anti-Hermitian part of the embedding self-energy.

The substrate local DoS (LDoS) $A_{\text{subs}}(\mathbf{r}, \omega)$ is determined by the substrate Green's function $\mathcal{G}(\mathbf{r}, \mathbf{r}', \omega)$ as $A_{\text{subs}}(\mathbf{r}, \omega) = -\text{Im}\mathcal{G}(\mathbf{r}, \mathbf{r}', \omega)/\pi$. The effect of the Anderson impurities on the substrate is expressed in a closed form[2] by a $T$-matrix relation. Assuming point-like impurities at locations $\mathbf{R}_i$ and a homogeneous substrate, we get

$$\mathcal{G}(\mathbf{r}, \mathbf{r}'; \omega) = \mathcal{G}^{(0)}(\mathbf{r} - \mathbf{r}'; \omega) \\ + \gamma \sum_{ij} \mathcal{G}^{(0)}(\mathbf{r} - \mathbf{R}_i; \omega) G_{ij}(\omega) \mathcal{G}^{(0)}(\mathbf{R}_j - \mathbf{r}'; \omega). \quad (5)$$

In the last equation, the Green's function of the unperturbed substrate is denoted with a superscript (0) and $G_{ij}(\omega)$ is the Green's function of the impurities.

In two dimensions, $\mathcal{G}^{(0)}(\mathbf{r} - \mathbf{r}'; E_F) = \frac{i}{2} H_0^{(2)}(k_F|\mathbf{r} - \mathbf{r}'|)$ (Hankel function). The first term in the right-hand side of Eq. (5) gives just the constant DoS and we discard it. The second term provides the change of the LDoS due to impurities, $\delta A_{\text{subs}}(\mathbf{r}, \omega)$. It reflects the propagation of the non-local Kondo resonance in the substrate. Finally, $\gamma$ is a constant proportional to $\Gamma_0$.

We use the latter approach to calculate the LDoS in long chains, as it captures well the interference effects at the length scales comparable with $\lambda_F = 2\pi/k_F$. However, in the two-impurity systems, we want to describe features at smaller length scales ($\approx .5a$, impurity spacing), where atomistic details matter. Accordingly, for the real space maps, we add to the homogeneous electron gas exponential wave functions $\varphi_i(\mathbf{r})$ to calculate the LDoS of the Anderson impurities $A_{\text{imp}}(\mathbf{r}; \omega)$ from

the impurity Green's function

$$A_{\text{imp}}(\mathbf{r}; \omega) = -\frac{1}{\pi} \text{Im} \sum_{ij} G_{ij}(\omega) \varphi_i(\mathbf{r}) \varphi_j(\mathbf{r}). \quad (6)$$

We use the wave functions $\varphi_i(\mathbf{r}) = C \exp(-|\mathbf{R}_i - \mathbf{r}|/\alpha)$ with radius $\alpha = 1.5$ Å. The two orbital centres are separated by $|\mathbf{R}_1 - \mathbf{R}_2| = 2.6a$. For such small length scales ($\ll \lambda_F$), we neglect the spatial dependence of $\Gamma_{ij}$, and parametrize its matrix elements as given in the main text with the following frequency dependence:

$$\Gamma_{ij}(\omega) = \Gamma_0 \exp(-(\omega/2E_0)^2). \quad (7)$$

We note that the actual Co d-orbitals are quite localized (orbital radius $\approx 0.5$ Å). Therefore, the $\varphi_i$ is to be viewed rather as a superposition of non-magnetic local states in the vicinity of each Co. These atomistic details become less important as $L$ approaches $\lambda_F$ and the description by means of the homogeneous electron gas (cf. Eq. (5)) is more appropriate.

We choose parameters that fall well within the Kondo regime in all cases: $U = \infty$, $\epsilon_d = -0.4$, $E_0 = 4$ and temperature $k_B T = 10^{-4}$ (all in eV). We take $\Gamma_0 = 0.06$ and 0.01 eV for long chains and dimers, respectively. The qualitative features of the local DoS are unchanged as long as we stay in the Kondo regime.

## Data availability
The spectroscopic data and microscopy images in binary format that support the findings of this study are available in DIGITAL.CSIC with the following persistent identifier: http://hdl.handle.net/10261/178494. Additional information can be obtained from the authors.

## Code availability
Computer code used to solve the Anderson Hamiltonian and generate the profiles and images of the spectral functions can be shared by the authors upon request.

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

## Acknowledgements

Financial support was provided by the Spanish Plan Nacional de I+D+i (grants MAT 2013-46593-C6-3-P, MAT2016-78293-C6-6-R, MAT2015-66888-C3-2-R, and FIS2015-64886-C5-3-P), Charles University (programme PRIMUS/Sci/09) and the European Union through programmes Interreg-POCTEFA (grant TNSI/EFA194/16) and H2020-EINFRA-5-2015 MaX Center of Excellence (grant no. 676598). M.M.-L., M.P., and D.S. acknowledge the use of SAI at Universidad de Zaragoza. R.R. acknowledges The Severo Ochoa Centers of Excellence Program (grant no. SEV-2017-0706) and Generalitat de Catalunya (grant no. 2017SGR1506 and the CERCA Programme).

## Author contributions

D.S. and M.M.-L. conceived the project. M.M.-L., M.P., J.I.P., and D.S. performed the experiments and analyzed the data. R.K. developed the MIAM model and carried out the associated calculations. N.L. and R.R. performed the DFT and the MIAM model calculations for dimers. All authors discussed the results and participated in the drafting of the paper. D.S., R.K, M.M.-L., and M.R.I. wrote the manuscript.

## Additional information

**Competing interests:** The authors declare no competing interests.

