## [Peer Review File · Nature Communications]

Reviewers' comments:

Reviewer #1 (Remarks to the Author):

In my previous report, I have made the point that this work is of good quality and represents an interesting novel direction in understanding the Kondo lattice physics using atomic manipulation. In the meanwhile, I have also raised the questions concerning technical and conceptual advances of the work, as well as the possibility of interpreting their data by spatial interference. The authors have made great efforts to clarify these issues. In particular, they have provided further discussions to exclude the effects of spatial interference. While I cannot personally confirm the advantage of their method, I think these issues have been noted in the revised manuscript to be examined by more experts. I agree with the authors that the finite size system may not be in exact correspondence with real Kondo lattice materials. It is thus interesting to explore their similarity and differences in the future. In any case, I think that nonlocal hybridization should be an important issue in related studies. The revisions are satisfying and I recommend the work to be published in Nature Communications.

Reviewer #2 (Remarks to the Author):

Moro-Langares et al., Real Space manifestations of coherent screening in atomic scale Kondo lattices.

The manuscript has improved from the version I first reviewed. At that point I already had a positive evaluation of the manuscript, however with some questions regarding the fitting procedure. The authors have addressed my concerns in their rebuttal letter. Specifically, they have explained the role different tip apex conditions have on the dI/dV spectrum, which relates to the fitting of the data in Fig. 2B. The authors also clarified that the spectroscopic images are obtained in constant height mode, which is a difficult technical accomplishment for these kinds of STS experiments. I have a few comments below. If these addressed I believe the manuscript should be published in Nature Communications.

1 – I think the discussion about the effect of different tips on the spectrum is very important and should be added to the supplementary information. I believe the authors take a reasonable approach, to ignore features that seem tip dependent and focus on the features that seem to have a sample origin.

2 – However, I have some concerns about the way the logic the authors used in explaining the tip effects. I understand that the basic observation is that the -10 mV feature is dependent on poking the tip. However, two conclusions could be reached from the tip-dependence of the dI/dV . One is that “it is not an intrinsic feature of the Kondo resonance”, as the authors argue. Two, the modification of the tip apex geometry and/or orbital configuration can itself change the tunneling interference paths, resulting in different q values in the Fano shape, or even a breakdown of the Fano description (e.g. more than 2 channels). Thus, I recommend that the discussion about the tip should be in supplemental material, similar to the discussion and data shown in the rebuttal, but with addition of this second scenario and discussion.

3 - The authors also mention they could replace the data in Fig. 2B with another set without the -10

mV contribution, albeit noisier. Please add this to the supplemental or indicate where it can be found if already present?

4 – On the bottom of page 7, the authors mention that the intensity re-distribution towards the dimer center is more intense with sharper tips. How do the authors estimate the sharpness of the tip? From atomic corrugation, or some spectral feature? Please clarify.

Reviewer #3 (Remarks to the Author):

In this paper, the authors study chains of Kondo screened Co atoms on a Ag(111) surface. They observe that, for sufficiently close spacing between the atoms, the Kondo resonance transitions into a correlated state that is no longer centered on the atom locations. Based on existing knowledge of the system they argue that they should be in the limit where the coupling between atoms and substrate is dominating, such that RKKY interactions between atoms can be neglected. As such, the correlated state can only be due to interaction of the Kondo clouds, which means that there is strong indication that the resulting Kondo chains can be seen as finite-sized analogues of a heavy fermion liquid. If correct, this would be a significant step forward in the realization of Kondo lattices.

I reviewed this manuscript before for Nature Physics (as Reviewer #3). It is clear that, compared to the first version, the authors have taken care to address the questions of each of the reviewers in much detail. Likewise, they have considerably improved the manuscript to make it clearer and more balanced in its claims. Currently, the paper reads very well. Both experimental data and theoretical calculations are of good quality. Having read the responses of the authors to the referees, I find their argumentation convincing and realistic. I think the scope is fitting for Nature Communications, so I would support publication.

I have only some minor comments left on the manuscript.

1. While the authors were quite thorough in answering my question why the Ag(111) surface would be expected to give much stronger substrate coupling than other systems, I did not find much of this explanation back in the manuscript. I would like to see a little more justification on page 4, after the sentence “To this end, ... inter-site interactions”. In addition, there could be a reference to e.g. Wahl et al., PRL 2004, in which there is a table that lists Kondo temperatures for various systems including the one studied here.

2. With the data shown in Fig. 3c for the atom separation of 2a, it could be nice to determine the integral of the Kondo intensity over the scanned area, and compare it to the integral taken over the panel corresponding to e.g. separation 4a. This would serve as a way to quantify to what extent the Kondo resonance got redistributed, as the authors describe, or if to some extent it also vanished after bringing the atoms in close proximity. This is just a suggestion.

Response to reviewers' comments

Nature Communications manuscript NCOMMS-18-38226-T

We are very pleased that the revised manuscript has addressed all of the items raised by the reviewers. Furthermore, we greatly appreciate the reviewers' effort to analyse in depth and discuss our rebuttal letter, and their very strong support for publication of our manuscript in Nature Communications. In the following we provide our reply to the minor comments raised in the second referral, together with the changes implemented in the manuscript to respond to the concerns of reviewer #2 and #3.

Reviewer #2 (Remarks to the Author):

1 – I think the discussion about the effect of different tips on the spectrum is very important and should be added to the supplementary information. I believe the authors take a reasonable approach, to ignore features that seem tip dependent and focus on the features that seem to have a sample origin.

We have added Supplemental Fig S2 showing the impact of tip resonances near Fermi level on the Kondo spectra. We place this figure right after the description of how the fitting algorithm skips the data of bias regions where the tip background could dominate over the Kondo resonance (i.e., a region of $1.5\Gamma_0$ around ϵ_0).

2 – However, I have some concerns about the way the logic the authors used in explaining the tip effects. I understand that the basic observation is that the -10 mV feature is dependent on poking the tip. However, two conclusions could be reached from the tip-dependence of the dI/dV . One is that "it is not an intrinsic feature of the Kondo resonance", as the authors argue. Two, the modification of the tip apex geometry and/or orbital configuration can itself change the tunneling interference paths, resulting in different q values in the Fano shape, or even a breakdown of the Fano description (e.g. more than 2 channels). Thus, I recommend that the discussion about the tip should be in supplemental material, similar to the discussion and data shown in the rebuttal, but with addition of this second scenario and discussion.

This is a very sensible comment. In the manuscript, we focus on the most frequent found line shape of the Kondo resonance, following the same criterion as in ref. [1]. We noticed that the feature at -10 mV can be observed rather seldom, but it is reproducible. As is concluded by the reviewer, our experimental data unambiguously link this feature to a tip property rather to an intrinsic characteristic of the Kondo resonance of Co/Ag(111). This is clearly shown by our new supplemental Figure S2, which we bring about as "*examples of spurious tip contributions to the Kondo resonance of individual Co atoms, which are identified as replicas of the features found in Co dimers built by atomic manipulation.*" In the subsequent discussion of the figure, we point out that one possible origin of the tip feature could be "*the change of the tunnelling matrix elements connecting tip states with different orbital symmetry and sample electrons participating in the many-body KS* ". The latter case goes beyond the Fano or Frotta description of the Kondo resonance. In addition, it is not of fundamental relevance to our work, which deals with the spatial dependence of the Kondo

parameters exclusively ascribed to the Co/Ag(111) system. Consequently, we omit these kind of tip-related low bias features in our quantitative analysis.

3 - The authors also mention they could replace the data in Fig. 2B with another set without the -10 mV contribution, albeit noisier. Please add this to the supplemental or indicate where it can be found if already present?

For the sake of clarity, we have substituted the data in Fig 2b by an equivalent data set, although with lower resolution (1 mV rms modulation instead of 0.5 mV). The advantage of this alternative dataset is that the asymmetry of the peak due to the -10 mV feature is absent, while the enhancement of the Kondo amplitude at the dimer's centre is very clear. Despite skipping the spectra that misled the reviewer in the previous referral round, we still discuss the other possible features of the tip at low bias in Fig. S2 to help the readers reproduce our findings, and refer to this figure in the main text with the following sentence: *"Care must be taken that there are not tip resonances contributing to the ZBF which could affect the intrinsic values of the fitting parameters (see Supplementary Figure S2)"*.

4 – On the bottom of page 7, the authors mention that the intensity re-distribution towards the dimer center is more intense with sharper tips. How do the authors estimate the sharpness of the tip? From atomic corrugation, or some spectral feature? Please clarify.

Yes, we estimate the sharpness from the corrugation in constant height current profiles at V_{ref} in the dimer at $d=2a$ using always the same scanning speed and tip-sample distance. As exemplified by right panel above with vertical profiles of the current images to the left, blunt tips typically show a dip of less than 10 pA in the middle of the dimers, while sharp tips show dips of 15 pA or larger. The experimentally measured A_k map in the left panel exhibits a more intense distribution towards the centre for the sharp tip, contrary to what would be expected if the tip convolution were responsible for the observed A_k redistribution in dimers (Fig. 3 of the manuscript). We have added in page seven the definition of sharpness in the sentence *"(sharpness is quantified by the corrugation of the tunneling current profiles at V_{ref} in dimers at $d=2a$)"*

Reviewer #3 (Remarks to the Author):

I have only some minor comments left on the manuscript.

1. While the authors were quite thorough in answering my question why the Ag(111) surface would be expected to give much stronger substrate coupling than other systems, I did not find much of this explanation back in the manuscript. I would like to see a little more justification on page 4, after the sentence "To this end, ... inter-site interactions". In addition, there could be a reference to e.g. Wahl et al., PRL 2004, in which there is a table that lists Kondo temperatures for various systems including the one studied here.

Following the reviewer's suggestion, we have slightly modified the end of the introductory section to highlight the differences between the (111) faces of all three noble metals Ag, Cu and Au studied in Wahl et. al 2004 (new reference 19 in the revised manuscript). In order not to alter the structure of the manuscript, we have postponed the discussion of its implications on a possible RKKY coupling to the section where we deliver the interaction between two Co atoms.

2. With the data shown in Fig. 3c for the atom separation of 2a, it could be nice to determine the integral of the Kondo intensity over the scanned area, and compare it to the integral taken over the panel corresponding to e.g. separation 4a. This would serve as a way to quantify to what extent the Kondo resonance got redistributed, as the authors describe, or if to some extent it also vanished after bringing the atoms in close proximity. This is just a suggestion.

Whilst in the case of a single atom the spatial integral of the Kondo intensity can be associated with the fraction of conduction electrons coupled to the Kondo resonance, we are unsure about its physical meaning in the case of a Kondo lattice. We thank the reviewer's suggestion but we would prefer not to introduce at this stage more speculative arguments without proper theoretical support. Since the raw data would be available at an open digital repository, we hope that our work triggers further theoretical efforts in the direction of multichannel screening of the finite Co lattices or the dynamic mean field approach to the Anderson Hamiltonian.